# Prevalence of *RAF1* Aberrations in Metastatic Cancer Patients: Real-World Data

**DOI:** 10.3390/biomedicines11123264

**Published:** 2023-12-09

**Authors:** Sung Hee Lim, Jaeyun Jung, Jung Young Hong, Seung Tae Kim, Se Hoon Park, Joon Oh Park, Kyoung-Mee Kim, Jeeyun Lee

**Affiliations:** 1Samsung Medical Center, Division of Hematology-Oncology, Department of Medicine, Sungkyunkwan University School of Medicine, Seoul 06351, Republic of Korea; polluxis82@gmail.com (S.H.L.); flandus@naver.com (J.J.); jungyong.hong@samsung.com (J.Y.H.); seungtae1.kim@samsung.com (S.T.K.); sh1767.park@samsung.com (S.H.P.); joonoh.park@samsung.com (J.O.P.); 2Experimental Therapeutics Development Center, Samsung Medical Center, Seoul 06351, Republic of Korea; 3Samsung Medical Center, Department of Pathology and Translational Genomics, Sungkyunkwan University School of Medicine, Seoul 06351, Republic of Korea; kim7353.kim@samsung.com

**Keywords:** RAF1 aberration, amplification, fusion, single-nucleotide variants, RAF inhibitor

## Abstract

Purpose: Therapeutic targeting of RAF1 is a promising cancer treatment, but the relationship between clinical features and RAF1 aberrations in terms of the MAPK signaling pathway is poorly understood in various solid tumors. Methods: Between October 2019 and June 2023 at Samsung Medical Center, 3895 patients with metastatic solid cancers underwent next-generation sequencing (NGS) using TruSight Oncology 500 (TSO500) assays as routine clinical practice. We surveyed the incidence of RAF1 aberrations including mutations (single-nucleotide variants [SNVs]), amplifications (copy number variation), and fusions. Results: Among the 3895 metastatic cancer patients, 77 (2.0%) exhibited *RAF1* aberrations. Of these 77 patients, 44 (1.1%) had *RAF1* mutations (SNV), 25 (0.6%) had *RAF1* amplifications, and 10 (0.3%) had *RAF1* fusions. Among the 10 patients with *RAF1* fusions, concurrent *RAF1* amplifications and *RAF1* mutations were detected in one patient each. The most common tumor types were bladder cancer (11.5%), followed by ampulla of Vater (AoV) cancer (5.3%), melanoma (3.0%), gallbladder (GB) cancer (2.6%), and gastric (2.3%) cancer. Microsatellite instability high (MSI-H) tumors were observed in five of 76 patients (6.6%) with *RAF1* aberrations, while MSI-H tumors were found in only 2.1% of patients with wild-type *RAF1* cancers (*p* < 0.0001). Conclusion: We demonstrated that approximately 2.0% of patients with metastatic solid cancers have *RAF1* aberrations according to NGS of tumor specimens.

## 1. Introduction

The RAF family of protein kinases, which includes ARAF, BRAF, and RAF1 (CRAF), comprises RAS-activated enzymes that initiate signaling through the MAPK cascade to control cellular proliferation, differentiation, and survival. The RAF family plays a pivotal role in transducing signals from RAS to downstream kinases, mitogen-activated protein kinase (MAPK), and extracellular signal–regulated kinase (ERK) kinase (MEK1/2) and ERK1/2 [1,2]. Well-known *BRAF* mutations have been reported in up to 20% of all types of cancers [3,4], and *BRAF V600E* targeting agents such as dabrafenib, vemurafenib, and encorafenib are used to treat melanoma, lung cancer, and colorectal cancer [5,6,7,8,9]. *BRAF* fusions are reported in 3% (14/531) of melanomas, 2% (3/1062) of gliomas, and approximately 1% of non-small cell lung carcinomas (NSCLCs) and colorectal cancers [10].

Regarding *RAF1, RAF1* mutations are very rare in contrast to *BRAF* mutations, and it has yet to be determined whether *RAF1* mutations constitute oncogenic drivers in human cancers. However, a previous in vitro study confirmed the oncogenic potential of *CRAF-S257L* and *CRAF-S259A* as well as the sensitivity of these mutants to RAF inhibition [11]. Recurrent rearrangements in *RAF1*, which are functionally similar to *BRAF* fusions, have been found to occur in advanced prostate cancers, gastric cancers, melanomas [12,13,14], and juvenile pilocytic astrocytomas [15].

In addition, amplification of the *RAF1* gene has been observed in urothelial cancers, and *RAF1* amplification drives the activation of MAPK signaling and exhibits a luminal gene expression pattern [16].

Emerging research on targeting *RAF1*-mediated signaling and the development of pan-RAF inhibitors are underway. Given their rarity, little is known about the overall incidence of *RAF1* aberrations in various solid tumors, and the significance of *RAF1* aberrations especially fusions and amplifications in clinical outcomes also remains unclear.

Given the challenges of the therapeutic approach to *RAF1* in oncology patients, we analyzed the incidence of *RAF1* mutations, amplifications, and fusions in 3895 patients with solid cancers on the basis of clinical sequencing.

## 2. Materials and Methods

### 2.1. Patient Enrollment

The collection of specimens and associated clinical data used in this study was approved by the Institutional Review Board of Samsung Medical Center (IRB# 2021-09-052). All patients who participated in this study provided written informed consent prior to enrollment and specimen collection. This study was performed in accordance with the principles of the Helsinki Declaration and the Korean Good Clinical Practice guidelines.

### 2.2. DNA Extraction

Tumor regions were micro-dissected for most tumor tissues, except for the samples used for genomic DNA extraction. Genomic DNA was isolated from formalin-fixed paraffin-embedded (FFPE) tissue fragments and purified using the AllPrep DNA/RNA FFPE Kit (Qiagen, Venlo, Netherlands). DNA concentrations were measured using a Qubit dsDNA HS assay kit (Thermo Fisher Scientific, Waltham, MA, USA), and 40 ng of DNA was used as the input for library preparation. The DNA integrity number, which is a measure of the DNA fragment size and consequently DNA quality, was determined using the Genomic DNA ScreenTape assay on an Agilent 2200 TapeStation System (Agilent Technologies, Santa Clara, CA, USA).

### 2.3. Library Preparation and Data Analysis

A DNA library was prepared using a hybrid capture-based TruSight Oncology 500 DNA/RNA NextSeq Kit following the manufacturer’s protocol. During library preparation, enrichment chemistry was optimized to capture nucleic acid targets from FFPE tissues. Unique molecular identifiers (UMIs) were used for TruSight Oncology 500 (TSO 500) analysis to determine the unique coverage at each position and reduce any background noise caused by sequencing and deamination artifacts in FFPE samples. During DNA library preparation, variants with low variant allele frequencies (VAFs) were detected while simultaneously suppressing errors, thereby providing high specificity.

Sequence data were analyzed for clinically relevant classes of genomic alterations, including SNVs and small insertions and deletions (indels), CNVs, and rearrangements/fusions. The results of SNVs and small indels with a variant allele frequency (VAF) of less than 2% were excluded. Average copy number variations of greater than 4 were considered gains, and those of less than one were considered losses. Only gains (amplifications) were analyzed in the TSO 500-CNV analysis, and RNA translocation-supporting reads of more than 4 to 12 were considered as translocations, which was dependent on the quality of the samples. Data outputs exported from the TSO 500 pipeline (Illumina, San Diego, CA, USA, Local App version 1.3.0.39) were annotated using the Ensembl Variant Effect Predictor (VEP) Annotation Engine with information from databases such as dbSNP, gnomAD genome and exome, 1000 Genomes, ClinVar, COSMIC, RefSeq, and Ensembl. The processed genomic changes were categorized according to a 4-tier system proposed by the American Society of Clinical Oncology/College of American Pathologists and annotated with proper references. The TSO 500 pipeline (Illumina, San Diego, CA, USA, Local App version 1.3.0.39) was used to determine the TMB and microsatellite instability (MSI) statuses. TMB was calculated by (1) excluding any variant with an observed allele count ≥ 10 in any of the gnomAD exome and genome and 1000 Genomes databases; including (2) variants in the coding region (RefSeq Cds), (3) variants with a frequency of ≥5%, (4) variants with coverage of ≥50×, (5) SNVs and indels; and excluding (6) nonsynonymous and synonymous variants. The effective panel size for TMB was the total coding region with coverage of >50×. MSI was calculated from the microsatellite sites according to the evidence of instability relative to a set of baseline normal samples based on information entropy metrics. The percentage of unstable MSI sites out of the total assessed MSI sites was reported as a sample-level microsatellite score.

### 2.4. Statistical Analysis

The data are presented as the mean ± SD. All statistical analyses were performed using R (Ver. 3.4), R studio (https://www.rstudio.com/ accessed on 1 January 2019), and GraphPad Prism 8.0 (GraphPad Software, San Diego, CA, USA; http://www.graphpad.com/ accessed on 1 January 2019). Statistical significance was set at *p* < 0.05. All statistical tests were two-sided.

## 3. Results

### 3.1. Patient Characteristics

Samples from a total of 3895 cancer patients were assessed by next-generation sequencing including 523 cancer genes (TSO500; Illumina) as routine clinical practice at Samsung Medical Center between October 2019 and June 2023. The most common tumor types were colorectal cancer (CRC) (*n* = 1350, 34.7%), gastric cancer (GC) (*n* = 920, 23.6%), cholangiocarcinoma (CCC) (*n* = 332, 8.5%), and sarcoma (*n* = 282, 7.2%) (Figure 1A). The tumor specimens from a total of 77 patients (2.0%) had *RAF1* aberrations. Of these 77 patients, 44 (1.1%) had *RAF1* mutations (SNV), 25 (0.6%) had *RAF1* amplifications, and 10 (0.3%) had *RAF1* fusions. Among the 10 patients with *RAF1* fusions, concurrent *RAF1* amplifications and *RAF1* mutations were identified in one each (Figure 1B).

Next, we investigated the prevalence of *RAF1* aberrations in each cancer type. The percentage of patients with *RAF1* aberrations was the highest in those with bladder cancer (11.5%), followed by ampulla of Vater (AoV) cancer (5.3%), melanoma (3.0%), gallbladder (GB) cancer (2.6%), and gastric cancer (2.3%) (Figure 1C). No significant difference in the tumor mutational burden (TMB) score or the PD-L1 combined positive score (CPS) was observed between patients with and without *RAF1* aberrations (Figure 1D,E). The median tumor mutational burden (TMB) score was 7.1 Muts/Mb in patients with *RAF1* aberrations compared to 5.5 Muts/Mb in those with wild-type *RAF1* (Figure 1D). In patients with RAF1 aberrations, the median PD-L1 (CPS) score was observed to be 5.0, whereas in those with wild-type RAF1, the median PD-L1 (CPS) score was 4.0 (Figure 1E). MSI-H tumors were observed in 5 out of 76 patients (6.6%) with *RAF1* aberrations, while MSI-H tumors were found only in 2.1% of wild-type *RAF1* cancer patients (*p* < 0.0001) (Figure 1F). At the time of diagnosis of metastatic disease in all patients, formalin-fixed paraffin-embedded tissue specimens were subjected to NGS. All software tools were used according to the Illumina “TruSight Oncology 500 v2.0 Local App” protocol. DNA alignment was performed using the BWA-MEM (https://bio-bwa.sourceforge.net/ accessed on 1 January 2019), CNV calling was conducted with CRAFT (https://support.illumina.com/help/BS_App_TruSigntTumor170_OLH_1000000028435/Content/Source/Informatics/CopyNumberVariantCaller_CRAFT.htm accessed on 1 January 2019), SNV calling was completed with Pisces (https://github.com/Illumina/Pisces accessed on 1 January 2019), fusion calling was carried out with Manta, annotation was performed with Nilrvana (https://illumina.github.io/NirvanaDocumentation/ accessed on 1 January 2019), TMB calculation was completed with TmbRaider, and MSI assessment was achieved with Hubble (Illumina).

### 3.2. RAF1 Amplification (CNV)

Of the 77 patients with RAF1 aberrations, 25 patients (32.5%) had RAF1 copy number variations (CNVs) in their tumor specimens. The most prevalent tumor types were bladder cancer (*n* = 11, 44%), followed by GC (*n* = 6, 24%), and melanoma (*n* = 4, 16.0%). *RAF1* amplifications were also observed in two patients each with CRC (*n* = 2, 8.0%) and sarcoma (*n* = 2, 8.0%) (Figure 2A). The degree of *RAF1* amplification ranged from 4.1 to 52.5 (median: 6.0) (Figure 2C). Of note, over 90% of the patients had fewer than 15 copy number of *RAF1* amplifications (x < 5; *n* = 6, 24%, 5 ≤ x < 10: *n* = 10, 40%, 10 ≤ x < 15: *n* = 7, 28%). The median value of the copy number was the highest in melanomas (10.5), while it was the lowest in GCs (5.25) (Figure 2B). There were no correlations between copy numbers and TMB scores.

Next, we evaluated *RAF1* amplification in correlation with the TMB status (≥ 10 mutations/Mb vs. < 10 mutations/Mb), MSI status (microsatellite stable [MSS] vs. MSI-high), and PD-L1 combined positive scores (CPSs) (CPS 0 v ≥ 1). We determined that four of the 25 patients had concurrent high TMB scores (Figure 2D). Three patients with *RAF1* amplifications exhibited positive PD-L1 CPSs, and all *RAF1* amplification tumors were MSS. Of note, the most common concomitant genetic aberration was *TP53* mutation, which was observed in 64% of all 25 patients. Following the *TP53* gene, *NOTCH3* (*n* = 7, 25%), *HIST1H1C* (*n* = 7, 25%), and *ATM* (*n* = 6, 24%) were the most frequently mutated genes in *RAF1*-amplified patients (Figure 2E).

### 3.3. RAF1 Mutation (Single-Nucleotide Variation)

In total, 44 patients (57.1%) had *RAF1* mutations among the 77 patients with *RAF1* aberrations. The most common cancer types were CRC (*n* = 18, 40.9%), GC (*n* = 11, 25.0%), and bladder cancer (*n* = 4, 9.1%) (Figure 3A). The pattern of nucleotide change varied depending on the cancer type—comparatively, hepatocellular carcinoma (HCC), bladder cancer, CRC, and GC exhibited a high proportion of C to T changes (Figure 3B). SNVs were identified at 40 distinct sites within the RAF1 gene, with the most frequent SNV observed at the Ala529 site (*n* = 3).

This was followed by SNVs at the Ser257, Ser259, Leu476, and Lsp486 sites, each with two occurrences. Both the mutations at Ala529 and Leu476 were identified in colorectal cancer, while those at Ser257 were exclusively found in gastric cancer (Figure 3C). Mutations in the *TP53*, *SPEN*, and *ARID1B* genes most frequently co-occurred with *RAF1* mutations (Figure 3D). On analyzing mutation types, the most common type was missense mutation (*n* = 461, 87.5%). Ten patients had in-frame insertions, and 17 patients had in-frame deletions (Figure 3D). MSI-H tumors were confirmed in four patients, and two of these patients were diagnosed with CRC, one with GC, and one with pancreatic cancer. PD-L1 positivity was found in the tumors of 12 out of 14 patients for whom PD-L1 assessments were available.

### 3.4. RAF1 Fusions

Of the 77 patients with *RAF1* aberrations, 10 patients (13.0%) had *RAF1* fusions in their tumor specimens. However, there was one patient who had four fusions and another patient with three, bringing the total number of fusions to 15. Of the 10 patients, 4 (40%) were diagnosed with GC, 2 with melanoma, 2 with pancreatic cancer, 1 with sarcoma, and 1 with GB cancer (Figure 4A).

Various fusion partners were observed (Figure 4B–D), and in GC patients, the KRT8, LSAMP, TMEM40, and TAMM41 genes were identified. Among the 10 RAF1 fusion (+) patients, one patient with pancreatic cancer had an MSI-H tumor and high TMB. When we assessed the landscape of mutations in RAF1 fusion patients, mutations in TP53, SETBP1, and TET2 genes were most frequent. Out of the 10 individuals with fusions, one did not present mutations in the top 17 genes and was therefore excluded from the landscape analysis (Figure 4D). Of note, *RAF1* mutation and amplification were detected simultaneously with *RAF1* fusion in one of each. The detailed fusion partners are outlined in Table 1. Except for TMEM40, all other fusion partner genes in this study are reported for the first time.

## 4. Discussion

This study represents a large-scale real-world data analysis of RAF1 aberrations including amplifications, fusions, and SNVs in various solid cancers. Overall, RAF1 aberrations were observed in the tumors of 77 patients (2.0%) among a total of 3895 patients whose tumor specimens were subjected to NGS. RAF1 mutations represented 57.1% of all RAF1 aberrations, amplifications accounted for 32.5%, and RAF1 fusions were observed in 13.0%. Of note, there was one patient with concurrent RAF1 amplification and RAF1 fusion and one patient with concurrent RAF1 fusion and RAF1 amplification. The frequency of MSI-H tumors was significantly higher in patients with RAF1 aberrations compared to those with RAF1 wild-type cancers (6.6% vs. 2.1%, *p* < 0.0001). In particular, MSI-H tumors were not found in RAF1-amplified cancers and were only identified in RAF1-mutated or RAF1-fusion cancers.

Our study represents the largest number of various cases of RAF1 gene aberrations described to date. Although RAF1 aberrations are infrequent in advanced solid cancers, RAF1 fusions have been previously identified in several solid tumors especially pediatric brain tumors and pancreatic acinar cell carcinomas [14,17]. A relatively high incidence of RAF1 gene rearrangements of 14.3–18.5% has been reported in pancreatic acinar cell carcinomas [14] and at a frequency of 0.6% (40/7119) in melanoma patients [18]. The prevalence of BRAF fusions has been reported to be approximately 0.3% in samples analyzed with previous comprehensive genomic profiling (0.3%, 55/20,537) [10] and Memorial Sloan Kettering (MSK) Impact testing (0.3%, 33/10,945) [19]. In the present study, RAF1 fusions involving the intact and in-frame RAF1 kinase domain were observed in 0.3% of all samples analyzed, and we observed 10 cases of RAF1 fusion, and these were composed of all different fusion partners: KRT8, TMEM40, LSAMP, and TAMM41 in gastric cancer; VPRBP in gallbladder cancer; TDRD10, IL6R, SHE, and SLC25A20 in melanoma; CACNA2D3 and PFKFB4 in pancreatic cancer; and APPL2 in sarcoma.

RAF1 fusions aberrantly activate the MAPK signaling pathways and additionally activate the phosphoinositide-3 kinase/mammalian target of rapamycin (PI3K/mTOR). Therefore, unlike BRAF fusions, tumors with RAF1 fusions do not respond to RAF inhibitors [17,20,21]. Previously, type-II BRAF inhibitors have shown preclinical activity inhibiting BRAF V600 mutations, BRAF fusions, and RAF1 [22,23]. Pan-RAF inhibitors as well as newer RAF-directed agents with novel mechanisms of action preferentially targeting RAF1-fusion or RAF1-amplified tumors are in development.

Bladder cancer was the most prevalent tumor having RAF1 amplifications with a frequency of 8.4% (11/131), which was slightly less frequent than a previous reported study and TCGA data. Bekele at al. demonstrated that RAF1 inhibition, with pan-RAF inhibitors and the combination of RAF1 inhibition with MEK inhibition, were efficacious in preclinical models harboring RAF1 amplifications [16]. Unlike BRAFV600E function, BRAF inhibitors preferentially bind and inhibit monomeric RAF. Since most RAF1 aberrations activate the MAPK pathway through dimerization, an alternative strategy for targeting RAF1 aberrations is required. Various RAF inhibitors with distinct mechanisms of action are now being tested in patients with tumor MAPK pathway alterations [24].

Among all patients with RAF1 aberrations, RAF1 CNVs were identified in the largest proportion (44 patients), but little is known about the clinical significance of RAF1 mutations. In lung adenocarcinomas with KRAS mutations, RAF1 ablation in tumors leads to significant regressions including some complete regressions [25]. In addition, certain mutations of RAF1 lead to kinase-inactive RAF1 with no effect on MAPK signaling [26]. Kinase-independent functions of RAF1 blocking apoptosis have been reported, and this activity is believed to be mediated by the inactivation of the proapoptotic kinases ASK1 and MST2 [27].

How to therapeutically target tumors driven by RAF aberrations, especially fusions and amplifications, has become an increasingly important question. Our report expands the landscape of oncogenic RAF1 aberrations in various solid cancers, and increasing the recognition of RAF1 aberrations in tumors will assist in further refining tumor classification and hopefully guide the management of patients with tumors bearing these alterations. Further research is warranted to analyze in-depth biological mechanisms and RAF1 aberrations in this patient group.

## 5. Conclusions

Our data revealed that when tumor specimens from patients with metastatic solid cancers are subjected to NGS, approximately 2.0% of these specimens exhibit *RAF1* aberrations. Overall, these findings identify a subset of molecularly defined RAF1-aberrated tumors that could be targeted using RAF1-directed therapy.

## Figures and Tables

**Figure 1 biomedicines-11-03264-f001:**
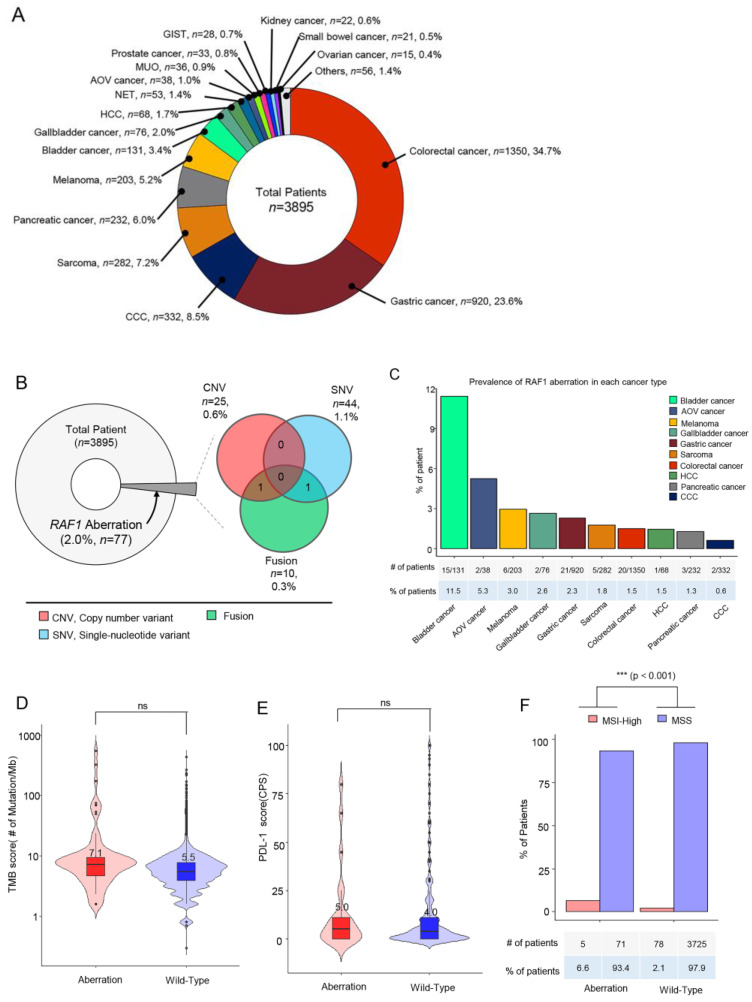
Overview of the enrolled cancer patients and the proportions of *RAF1* genetic variants. (**A**) Between October 2019 and June 2023, all patients with stage IV cancer at the Precision Oncology Clinic of Samsung Medical Center were screened for *RAF1* aberrations through next-generation sequencing using a panel targeting 500+ genes (TruSight Oncology Next Seq). A pie chart indicating the percentages of each type of cancer in a total of 3895 patients. (**B**) A pie chart representing the proportion of patients with any RAF1 aberration (left), and a Venn diagram showing the number and percentage of patients with RAF1 CNVs (amplifications), SNVs, and fusions. (**C**) The proportion of patients with RAF1 aberrations according to each cancer type. (**D**) The TMB scores between tumors with RAF1 aberrations and wild-type tumors. (**E**) The PD-L1 scores between tumors with RAF1 aberrations and wild-type tumors. ‘ns’ (**F**) The percentage of patients with MSI-H tumors between tumors with RAF1 aberrations and wild-type tumors. Statistical significance was assessed by the U-test. *p*-values < 0.05 were considered significant; (***) *p*-value < 0.001; ns = not significant.

**Figure 2 biomedicines-11-03264-f002:**
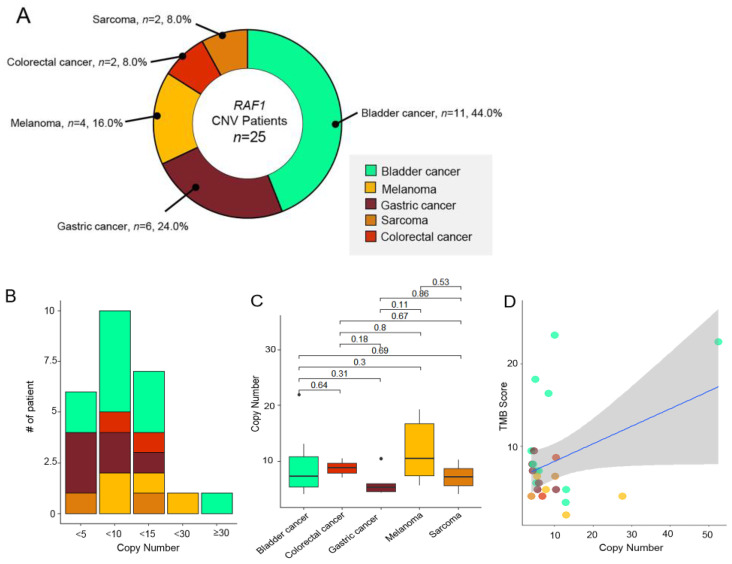
(**A**) Pie chart showing the distribution of the percentage of tumor types with RAF1 amplifications (*n* = 25): bladder cancer (*n* = 11, 44%), GC (*n* = 6, 24%), and melanoma (*n* = 4, 16.0%) in order of the most frequently observed tumor types. (**B**) Chart showing the number of patient incidences by RAF1 copy number range. (**C**) The range of copy number for each cancer type. The square point represented the mean value of the copy number. (**D**) No correlations between the RAF1 copy number and the TMB score. (**E**) Landscape of the patient’s genomic profiles. The first top panel: copy number of the *RAF1* gene. The second top panel: TMB score; middle: TMB, cancer type, sex, age, microsatellite instability, and PD-L1 status; and bottom: OncoPrint showing concurrent SNV genes in *RAF1*-amplified patients. Left: top gene list of the most frequently mutated genes and the percentage of the mutation in *RAF1*-amplified patients. CNV, copy number variation; GC, gastric cancer; CPS, combined positive score; IHC, immunohistochemistry; MSI, microsatellite instability; MSS, microsatellite stable; and TMB, tumor mutational burden.

**Figure 3 biomedicines-11-03264-f003:**
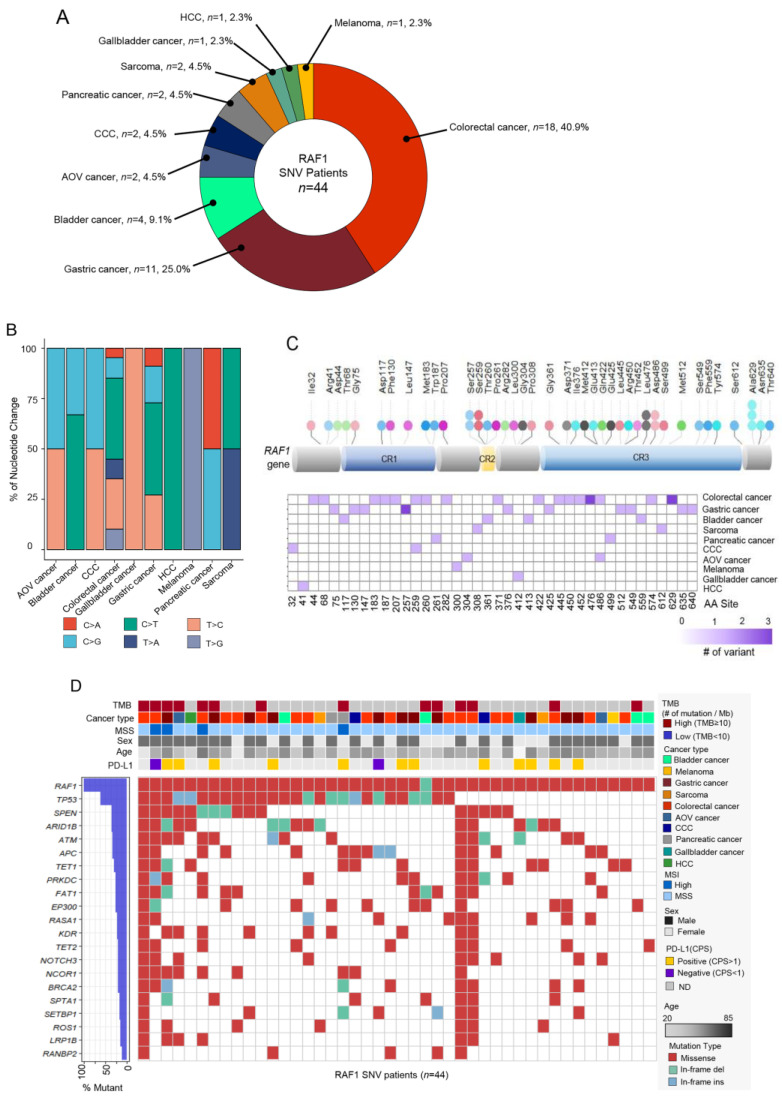
(**A**) Pie chart showing the cancer type of *RAF1*-SNV patients: CRC (*n* = 18, 40.9%), GC (*n* = 11, 25.0%), and bladder cancer (*n* = 4, 9.1%) in order of the most frequently observed tumor types. (**B**) Bar graph representing the proportion of each nucleotide change in various cancer types. (**C**) Lollipop plot showing the position and number of specific *RAF1* aberrations that occurred in the *RAF1* gene. The bar represents the structure of the *RAF1* gene. The length of the lollipop is proportional to the number of mutations. (**D**) Landscape about several clinical factors and OncoPrint corresponding to SNV mutations in other genes in *RAF1*-SNV patients. Top panel: TMB, cancer type, sex, age, MSI, and PD-L1 status; bottom panel: OncoPrint showing the SNV of other genes; and left panel: the percentage of the mutation in the total sample (*RAF1*-SNV patients; *n* = 44). SNV, single-nucleotide variant; AOV, ampulla of Vater; CCC, cholangiocarcinoma; HCC, hepatocellular carcinoma; TMB, tumor mutational burden; CPS, combined positive score; MSI, microsatellite instability; and MSS, microsatellite stable.

**Figure 4 biomedicines-11-03264-f004:**
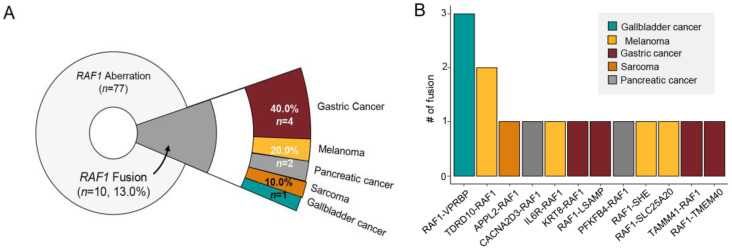
(**A**) Pie chart showing the distribution of cancer patient groups containing fused *RAF1* genes: GC (*n* = 4, 40%), melanoma (*n* = 2, 20%), and pancreatic cancer (*n* = 2, 20%) in order of the most frequently occurring tumor types. (**B**) The fusion number in each type of fused *RAF1* genes. Network diagram representing *RAF1* and fusion partner genes in GCs (**C**) and other cancers (**D**). (**E**) Landscape of the *RAF1*-fused patient’s genomic profile comprising the TMB score, cancer type, MSI status, sex, age, and PD-L1 (top panel). OncoPrint showing SNV of other genes (bottom panel); left panel: the percentage of mutations in the total sample (*RAF1*-fusion patients; *n* = 10).

**Table 1 biomedicines-11-03264-t001:** Detailed clinical information on patient-specific *RAF1* gene fusions.

Tumor Type	Fusion Frequency	Fusion Gene
Gastric cancer (*n* = 4)	1	*KRT8-RAF1*
	1	*RAF1-TMEM40*
	1	*RAF1-LSAMP*
	1	*TAMM41-RAF1*
Gallbladder cancer (*n* = 1)	3	*RAF1-VPRBP*
Melanoma (*n* = 2)	2	*RAF1-TDRD10*
	1	*IL6R-RAF1*
	1	*RAF1-SHE*
	1	*RAF1-SLC25A20*
Pancreatic cancer (*n* = 2)	1	*CACNA2D3-RAF1*
	1	*PFKFB4-RAF1*
Sarcoma (*n* = 1)	1	*APPL2-RAF1*

## Data Availability

All the data presented in this study are available upon request from the corresponding author. The NGS raw data of 77 patients were uploaded to ENA (European Nucleotide Archive).

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
