# Peer review of "Prevalence of RAF1 Aberrations in Metastatic Cancer Patients: Real-World Data"

_biomedicines, 2023, doi:10.3390/biomedicines11123264_

Round 1

Reviewer 1 Report

Comments and Suggestions for Authors

In the present manuscript, it reported that approximately 2.0% of 3,895 cancer patients have RAF1 aberrations by next-generation sequencing (NGS) using TruSight Oncology 500 (TSO500) assay. The manuscript is well-written. However, there are some points to correct. I recommend that this paper be accepted after minor revision.

1. In the legend of Figure 1, the font size was changed because there is a line break in the middle of the sentence.

2. In Materials and Methods, it should clarify the software for mapping and variant call and software version.

3. In line 9 on Page 5, correct Figure 2C to Figure 2B.

4. In Figure 4E, the color of MSS High is different. The color should be unified to either.

5. In line 46 on Page 11, there is an extra space in the sentence.

Author Response

We really appreciate your valuable comments about our study. Below we provide a point-to-point response to each of the comments, and we highlighted the revised parts of whole manuscript. 

1. In the legend of Figure 1, the font size was changed because there is a line break in the middle of the sentence.

: OK, we corrected the legend of Figure 1.

2. In Materials and Methods, it should clarify the software for mapping and variant call and software version.

: We added the detailed information of software tools using the analyses in the Material and Method section. 

All software tools were used according to Illumina's "TruSight Oncology 500 v2.0 Local App" protocol . The DNA alignment was performed using the BWA-MEM (https://bio-bwa.sourceforge.net/), CNV calling was conducted with CRAFT (https://support.illumina.com/help/BS_App_TruSightTumor170_OLH_1000000028435/Content/Source/Informatics/CopyNumberVariantCaller_CRAFT.htm), SNV calling with Pisces (https://github.com/Illumina/Pisces), fusion calling with Manta, annotation with Nilrvana (https://illumina.github.io/NirvanaDocumentation/), TMB calculation with TmbRaider, and MSI assessment with Hubble (Illumina).)

3. In line 9 on Page 5, correct Figure 2C to Figure 2B.

: We corrected it to Figure 2B.  

4. In Figure 4E, the color of MSS High is different. The color should be unified to either.

: Thank you for your comment on the details. The MSI-H color of figure 4E chart and index color have been unified. 

5. In line 46 on Page 11, there is an extra space in the sentence.

: We corrected it. 

Reviewer 2 Report

Comments and Suggestions for Authors

The authors surveyed the incidence of RAF1 aberration including mutation (single-nucleotide variant [SNV]), amplification (copy-number variation), and fusion in 3,895 patients with metastatic cancer patients received a next-generation sequencing (NGS) using TruSight Oncology 500 (TSO500) assay. Their results showed that when patients with metastatic solid cancer receive NGS test, approximately 2.0% have RAF1 aberrations in their tumor specimen. There are several problems:

1.      The authors need to upload the NGS data onto public available database.

2.      The authors only calculated very simple statistics. What conclusions can be drawn from each figure? For example, in Figure 2D, what did this figure indicate?

3.      Figure 2D and Figure 3D were very similar. Where the CNV and SNV expected to be similar?

4.      In Table 1, why the frequency was so low?

5.      The authors need to add more in-depth biological mechanism analysis, such as eQTL regulatory network.

Comments on the Quality of English Language

The authors surveyed the incidence of RAF1 aberration including mutation (single-nucleotide variant [SNV]), amplification (copy-number variation), and fusion in 3,895 patients with metastatic cancer patients received a next-generation sequencing (NGS) using TruSight Oncology 500 (TSO500) assay. Their results showed that when patients with metastatic solid cancer receive NGS test, approximately 2.0% have RAF1 aberrations in their tumor specimen. There are several problems:

1.      The authors need to upload the NGS data onto public available database.

2.      The authors only calculated very simple statistics. What conclusions can be drawn from each figure? For example, in Figure 2D, what did this figure indicate?

3.      Figure 2D and Figure 3D were very similar. Where the CNV and SNV expected to be similar?

4.      In Table 1, why the frequency was so low?

5.      The authors need to add more in-depth biological mechanism analysis, such as eQTL regulatory network.

Author Response

We really appreciate your valuable comments about our study. Below we provide a point-to-point response to each of the comments, and we highlighted the revised parts of whole manuscript. 

  1. The authors need to upload the NGS data onto public available database.

: We are fully willing to provide our NGS raw data if readers request it after the paper is published. Also we can provide it if necessary during paper review. 

  1. The authors only calculated very simple statistics. What conclusions can be drawn from each figure? For example, in Figure 2D, what did this figure indicate?

: Thank you for the comment, and we all agree with your comment.

In this study, considering the therapeutic approach for RAF1 aberrations in oncology patients is emerging, we tried to analyze the incidence of RAF1 aberrations in metastatic cancer patients. Among 3895 patients with metastatic cancer received NGS as routine practice, we found only 77 (2.0%) patients having RAF1 aberrations. We focused on describing the types and characteristics of rare RAF1 aberrations.

Figure 2E showed the landscape of RAF1-amplified patient’s genomic profiles.      

  1. Figure 2D and Figure 3D were very similar. Where the CNV and SNV expected to be similar?

: We did not expect the similar aspect of concomitant genetic aberration between patients with RAF1 amplification and RAF1 mutation. TP53 mutation was the most common co-occurred aberration in both patient groups. Except TP53 mutation, in Figure 2E, NOTCH3, HIST1H1C, and ATM were most frequently mutated gene in RAF1-amplified patients. In Figure 3D, SPEN and ARID1B gene were most frequently co-occurred with RAF1 mutation.

  1. In Table 1, why the frequency was so low?

: There were 10 patients with RAF1 fusion aberration, four with gastric cancer, two with melanoma, two with pancreatic cancer and one patient with sarcoma and GB cancer. Because one melanoma patient had four fusions and a GB cancer patient had three fusions, the total number of fusion frequency is counted as 15 in the Table 1. We added the number of patients of each cancer types in the Table 1.  

  1. The authors need to add more in-depth biological mechanism analysis, such as eQTL regulatory network.

: Thank you for the comment. As answered in comment 2, in this paper, we found that the overall incidence of RAF1 aberrations in patients with metastatic solid tumors was very low, and in particular, the frequency of RAF1 CNVs and fusions that could be targets of RAF1 treatment was even lower.

Further research is needed for in-depth biological mechanism analysis of these group of patients. We added your valuable comments as our study limitation in the Discussion section.

Round 2

Reviewer 2 Report

Comments and Suggestions for Authors

It is mandatory to upload the data and provide the accession number.

Author Response

It is mandatory to upload the data and provide the accession number.

--> We uploaded the NGS data of 77 patients with RAF1 aberrations to the submission site.

Please understand that NGS data for all 3895 patients cannot be shared publicly. 

Thank you. 

Round 3

Reviewer 2 Report

Comments and Suggestions for Authors

The authors can upload the data onto dbGAP like database, which requires application for accession. If the authors refuse to submit the data, I can't evaluate whether this study is true.

Author Response

According to reviewer's comment, we completed uploading NGS raw data (bam file) of 77 patients. The raw files were uploaded to ENA (European Neucleotide Archive) https://www. ebi.ac.uk/ena/submit/webin/.

Access number is PRJEB70258 (secondary accession: ERP155196). 

It took a long time and thank you for your patience. 

We really appreciate your comments about our study and wish our manuscript could be accepted for publication.